# Modified Vertical Parasagittal Sub-Insular Hemispherotomy—Case Series and Technical Note

**DOI:** 10.3390/brainsci13101395

**Published:** 2023-09-30

**Authors:** Nicole Del Gaudio, Susana Ferrao Santos, Christian Raftopoulos

**Affiliations:** 1Neurosurgery Department, University Hospital Saint Luc, Université Catholique de Louvain, Av. Hippocrate 10, 1200 Brussels, Belgium; nicole.delgaudio@student.uclouvain.be; 2Neurology Department, University Hospital Saint Luc, Université Catholique de Louvain, Av. Hippocrate 10, 1200 Brussels, Belgium; susana.ferrao@saintluc.uclouvain.be; 3Hospital Delta, CHIREC Boulevard du Triomphe 201, 1160 Brussels, Belgium

**Keywords:** drug-resistant epilepsy, hemispherotomy, surgical procedure

## Abstract

(1) Background: Hemispherotomy is the generally accepted treatment for hemispheric drug-resistant epilepsy (DRE). Lateral or vertical approaches are performed according to the surgeon’s preference. Multiple technical variations have been proposed since Delalande first described his vertical technique. We propose a sub-insular variation of the vertical parasagittal hemispherotomy (VPH) and describe our case series of patients operated on using this procedure. (2) Methods: Data from a continuous series of patients with hemispheric DRE who were operated on by the senior author (CR) using the modified sub-insular VPH technique were analyzed retrospectively. Pre-operative demographic and epilepsy characteristics, functional outcome, and surgical complications were extracted from medical charts. (3) Results: Twenty-five patients were operated on between August 2008 and August 2023; 23 have at least 3 months of follow-up. Of this group, 20 (86.9%) patients are seizure-free. Only two patients developed postoperative hydrocephalus (8.7%). All patients who were able to walk autonomously preoperatively and 20 (86.9%) of those with follow-up were able to walk without assistance. A total of 17 (74%) patients were able to perform adapted social activities at the latest follow-up. (4) Conclusions: Modified sub-insular VPH is a successful surgical technique for hemispheric DRE with seizure freedom rates similar to the largest series reported in the literature. Compared to other series, patients who were operated on with our modified technique had a lower rate of postoperative hydrocephalus and excellent long-term motor and cognitive outcomes.

## 1. Introduction

### 1.1. Historical Context

Patients with drug-resistant epilepsy (DRE) as a result of unilateral hemispheric pathology can benefit from surgical management, with a reported seizure freedom of 70–90% [1,2].

The classical anatomical hemispherectomy was first described by Dandy for glioma surgery [3] and involved resection of the entire cerebral hemisphere, leaving the basal ganglia in place. This technique was later used to treat epilepsy with 85% complete or near-complete seizure freedom [4]. Unfortunately, long-term follow-up showed neurological worsening in about 35% of patients. Autopsy reports revealed the appearance of delayed superficial cerebral hemosiderosis, characterized by meningeal inflammation with macrophages filled with hemosiderin, subpial necrotic lesions, and ependymitis [5,6,7].

Rasmussen was the first to propose a combination of resection and disconnection [5] for surgical treatment of hemispheric DRE. The purpose of his functional hemispherectomy was to reduce surgical morbidity [5,6,7] while maintaining the good seizure outcomes of anatomical hemispherectomy. His procedure involved a subtotal resection of the hemisphere, keeping the frontal and occipital lobes in place [1,5]. This new approach led to the introduction of hemispherotomy with even less resection and maximal disconnection [8]. 

Hemispherotomy can be described as a disconnection of the corona radiata and the internal capsule, the corpus callosum, the fornix, the intra-limbic and limbic gyri, the fronto-temporo-limbic connections, and the anterior commissure [1,9,10,11,12,13]. 

### 1.2. Technical Evolution

Several modifications of hemispherotomy have been described including anterior temporal lobectomy with peri-sylvian transcortical incision [14], using a supra-insular window [15,16], or a vertical approach through the central cortex [17,18]. The latter, the vertical parasagittal hemispherotomy (VPH), was described by Delalande and is characterized by a vertical disconnection with limited resection [17]. This method rapidly became the preferred method of performing hemispherotomy. 

### 1.3. Goal of this Study

To date, there is no definitive proof showing the superiority of one technique over another, either in terms of epileptic seizure freedom or complication rates (notably postoperative hydrocephalus, which implies additional surgery and shunt dependence) [1,19,20].

The purpose of this study was to report our experience with a modified sub-insular VPH technique, trying to reduce the complication rate (particularly the rate of shunt placement (15%)) and mortality (3.6%)) of classical hemispherotomy [17], while maintaining good epilepsy outcomes and preserving the lenticular nucleus (LN) as much as possible by performing a disconnection under the insular cortex (sub-insular).

## 2. Materials and Methods

### 2.1. Population and Data Collection 

All the patients had DRE, according to definitions from the international league against epilepsy (ILAE). Patients had a presurgical workup at the Saint-Luc University Hospital, which included video-EEG seizure recording in the epilepsy monitoring unit (EMU), high-resolution magnetic resonance imaging (MRI), fluorodeoxyglucose positron-emission tomography (FDG PET), and, if possible, a cognitive evaluation. Before surgery, all cases were discussed in our multidisciplinary meeting for DRE. Senior adult or pediatric neurology department staff members assessed postoperative follow-up. All relevant pre-, peri-, and postoperative data were collected retrospectively from medical charts. 

Epilepsy etiology was classified as congenital (cortical dysplasia, hemimegalencephaly, Sturge–Weber syndrome), acquired (ischemic or hemorrhagic stroke, postoperative, or post-traumatic encephalomalacia), and progressive (Rasmussen encephalitis).

### 2.2. Sub-Insular VPH Method Description (Figure 1)

The senior author (CR) performed all surgeries. Our first sub-insular VPH was conducted in August 2008 and included a series of modifications from the original Delalande VPH [17] with the aim of improving disconnection and preserving a maximum of lenticular connections. All surgeries were prepared on the BrainLab^®^ neuronavigation software (Brainlab AG, Feldkirchen, Germany) using MRI acquired just before the surgery. All volumes of interest were identified: Trolard vein (superior anastomotic vein), the targeted ventricle, the pericallosal and anterior cerebral arteries, the homolateral optic nerve, the great cerebral vein of Galen, the homolateral LN, and the amygdala. 

**Figure 1 brainsci-13-01395-f001:**
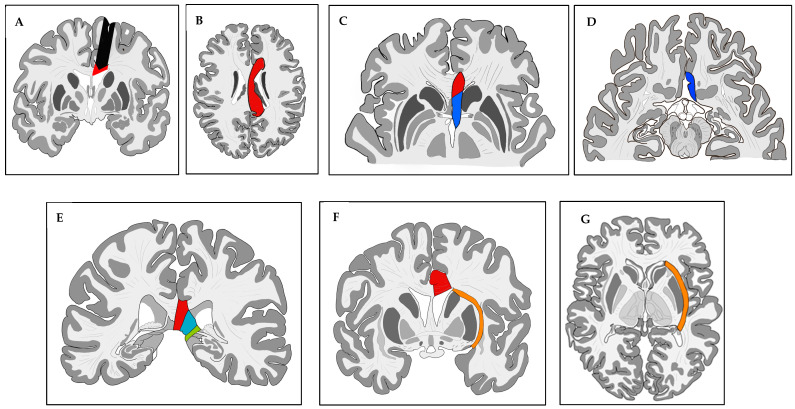
Major surgical steps of the modified sub-insular VPH. (**A**) Coronal section through the interventricular foramen showing the precentral parenchymal resection (black arrow) performed to enter the ventricular body and partial callosotomy (red). (**B**) Axial section through the corpus callosum showing the central callosotomy (red). (**C**,**D**) Axials sections through the anterior commissure and sub-callosal area showing the anterior corpus callosotomy (red) and sub-rostral resection (blue). (**E**) Coronal section through the ventricular trigone showing splenium (red) and ventricular trigone floor disconnection (with disconnection of the posterior column of fornix (light blue) and of the intralimbic and limbic gyri (green). (**F**,**G**) Coronal section through the anterior commissure and axial section through the striate body showing the subinsular trans-claustral disconnection (orange) and corpus callosotomy (red).

Surgery is always performed under general anesthesia. Our paramedian skin incision is centered slightly anterior to the coronal suture to avoid the Trolard vein. 

After dural opening with maximal preservation of the bridging veins, we perform, using the operative microscope, a minimal precentral parenchymal resection (pallium and body of corpus callosum; ≤20 mm by ≤10 mm). Once the ventricle body is opened and the interventricular foramen of Monro identified, we occlude it with a thin rectangular Gelfoam sponge (Pfizer Inc.^®^, New York, NY, USA) to avoid blood contamination of the distal ventricular system. Using the ultrasonic aspirator (CUSA Excel^®^ Integra LifeSciences^®^, Princeton, NJ, USA), we perform an anterior corpus callosotomy (genu and rostrum) followed by a sub-rostral resection of the posterior part of the gyrus rectus, of the cingulum, and of Brodmann area 25. Our next step consists of splenium disconnection down to the great cerebral vein of Galen. The ventricular trigone floor is then disconnected; during that step, we disconnect not only the crus (posterior column) of the fornix but also the intralimbic and limbic gyri to reach the posterior part of the ambient cistern. Once this step is completed, we reach the posterior part of the temporal horn and perform the posterior–anterior sub-insular trans-claustral disconnection, remaining as lateral as possible to the LN (Figure 2).

We then resect the piriform lobe and end our hemispherotomy by extending our disconnection along the Sylvian fissure until reaching the already resected subgenuorostrum area. At the end of the sub-insular VPH, we take time to achieve complete hemostasis before removing the Gelfoam plug from the foramen of Monro. A Gelfoam plug is placed in the cortical resection cavity and the dura, skull, and skin are closed as usual.

The patient usually remains in the intensive care unit for one day. Immediate postoperative imaging is not routinely performed, and after approximatively one week’s surveillance on the neuro-pediatric department, the patient is transferred to an associated institution (Centre Hospitalier Neurologique William Lennox, Ottignies-Louvain-la-Neuve, Belgium) for rehabilitation and follow-up. 

The principal differences between the Delalande and Raftopoulos techniques are shown in Table 1. 

### 2.3. Outcome 

The Engel classification was used to assess postoperative seizure outcomes [21].

### 2.4. Statistical Analysis 

Qualitative variables and quantitative variables are presented as numbers of patients and percentages with means, respectively. Statistical analysis was performed using GraphPad Prism 9. Results were considered significant when the *p*-value was less than or equal to 5% (*p* ≤ 0.05). A non-parametric Mann–Whitney test was conducted to compare the age variable in two populations: transfused and non-transfused patients. Data are presented as mean ± standard error of the mean (SEM). Fisher’s exact test was used to assess whether an acute postoperative seizure (APOS) complication influenced the Engel score.

### 2.5. Ethics 

This study was approved by the Clinical Research Ethics Board of the Cliniques Universitaires Saint-Luc (CUSL). 

## 3. Results

### 3.1. Demographic Data and Clinical Findings

We report on the first 25 consecutive patients with hemispheric DRE who had modified sub-insular VPH by the senior author (CR) at St Luc University Hospital. The mean age at seizure onset was 2.7 years (range: 0.0–9.6). The etiology was ischemic or hemorrhagic stroke in 13 patients (52%), Rasmussen syndrome in 4 (16%), cortical dysplasia in 2 (8%), hemimegalencephaly in 3 (12%), Sturge–Weber syndrome in 1 (4%), and gliosis after trauma or tumor resection in 2 (8%) (Table 2).

#### Medical History 

Four patients had had previous brain surgery. Patient 5 had had a decompressive craniectomy after trauma in another hospital. Patient 7 had had four epilepsy surgeries, with the most recent procedure being a peri-sylvian hemispherotomy and ventriculoperitoneal shunt (VPS) abroad without seizure improvement. Patient 11 had had drainage of a grade 4 intra-ventricular hemorrhage (HIV) with intraparenchymal hematoma (HIP). Patient 15 had already had a callosotomy by the senior author (CR) without seizure reduction. 

All patients had hemiparesis or hemiplegia before surgery without useful hand function. All patients had at least a moderate developmental delay.

### 3.2. Surgical Procedure and Postoperative Course (Table 3) 

The mean age at sub-insular VPH was 7.3 years (range: 0.16–22.1). Seventeen (68%) surgeries were performed on the right hemisphere. There were no intra-operative complications and no deaths. Five (20.8%) patients needed a blood transfusion during surgery or immediately afterward (average: 123 mL). The mean patient age was significantly lower in transfused than in non-transfused patients (Figure 3). 

**Table 3 brainsci-13-01395-t003:** Demographic data and outcome.

Case No	Sex	Age at (y)		Side	Etiology			Complications				2nd Surgery	FUp (y)	mRS	Sz Outcome (Engel)
		Onset	Surgery		Cong	Ac	Prog	BTF (mL)	HCP	APOS	AF				
1	F	6.6	6.8	R			+	0	-		+		14.91	1	I
2	M	7.4	8.9	R		+		0	-	+	-		13.59	1	I
3	F	3.2	5.7	R			+	100	-	+	+	+	11.77	5	IV
4	M	0	6.8	L		+		0	-		-		9.72	2	I
5	M	9.6	12.7	R		+		0	+		+		9.96	3	II
6	M	3.2	5.5	R	+			0	-		-		7.32	3	I
7	M	0.1	12.8	R	+			0	-		-		10.04	3	I
8	M	1	22.1	L		+		0	-	+	-		9.71	3	I
9	M	1.8	10	R		+		0	-		+		7.74	3	I
10	F	3	15.8	L		+		0	-		+		4.49	NA	II
11	F	0.3	9.7	R		+		0	-		-		7.6	3	I
12	F	1.5	4.9	L		+		180	-		+		7.02	2	I
13	M	3	5.5	L		+		0	-		+		6.86	3	I
14	F	3.7	4.2	R			+	0	+		-	+	6.2	2	I
15	M	1.3	6.9	R		+		0	-		-		4.31	3	I
16	F	6	7.4	R			+	0	-		-	+	5.15	2	I
17	F	4.6	5.5	R		+		0	-		+		4.6	3	I
18	M	0.3	1.5	L	+			90	-		-		4.95	3	I
19	M	0	1.6	R	+			0	-	+	+		4.93	4	I
20	F	3	4.8	L		+		+	-		+		4.82	4	I
21	F	0.17	2.38	R	+			0	-		-		3.95	2	I
22	M	0.75	7.19	R		+		0	-		-		3.33	3	I
23	F	5.98	10.51	L		+		0	-		+		0.32	3	I
24	M	0	0.16	R	+			100	-	+	-	+	0.16	NA	III
25	M	0.5	3.28	R		+		0	-	+	-		0.08	4	I
M/F	1.27														
Mean		2.68	7.3										6.5	2.8	
Range		9.6–0	22.1–0.16										14.9–0.08		

Cong, congenital; Ac, acquired; Prog, progressive; BTF, blood transfusion; HCP, hydrocephalus; APOS, acute postoperative seizure; AF, aseptic fever; FUp, follow-up; mRS, modified Rankin scale; Sz, seizure; NA, not available/not applicable; -, absent; +, present. Second surgeries (3, completion of VPH and KPS; 14, amygdalohippocampectomy; 16, amygdalohippocampectomy; 24, amygdalohippocampectomy and completion of callosotomy).

#### 3.2.1. Acute Postoperative Seizures 

Six (24%) patients had postoperative seizures (APOS) within the first week; however, four of them still achieved prolonged seizure freedom (Engel I). The two others (patients 3 and 24) needed a second surgical procedure to complete the hemispherotomy and were not seizure-free at the most recent follow-up appointment. There was no link between the presence of APOS and the risk of not being seizure-free (Figure 4). 

#### 3.2.2. Hydrocephalus and Shunting

No patient required a shunt during the first postoperative week. Two patients (9%) developed symptomatic delayed hydrocephalus requiring a shunt at one (patient 5) and three (patient 20) months postoperatively; the first patient needed a surgical revision three years after shunt placement for disconnection of the distal catheter, no other shunt complication occurred. Patient 3 had a cysto-peritoneal shunt placement during her second hemispherotomy procedure for a persistent subdural hygroma without hydrocephalus. Patient 15 needed a VPS for treatment of a recurrent pseudo-meningocele.

### 3.3. Seizure Outcome

Our mean follow-up period was 6.5 years (range: 0.08–11.7). Twenty-three patients had at least 3 months of follow-up; of those, 20 (86.9%) were Engel 1 at their most recent follow-up. Our results are comparable with those reported in the literature (Figure 5 and Table 4).

Patient 3, with Rasmussen encephalitis, was seizure-free for 10 months after the first VPH before recurrence of catastrophic status epilepticus. Postoperative MRI showed suspected persistence of a callosal connection. We performed a second surgery for completion of the hemispherotomy. Unfortunately, the patient’s seizures did not improve despite radiological confirmation of complete disconnection.Patient 5 developed epilepsy after severe head trauma that required decompressive craniectomy. He was seizure-free for two years after VPH and then presented recurrent spasms despite complete disconnection on MRI.Patient 10 still suffered from morpheic seizures after surgery, but her last video-EEG showed a bilateralization of the epileptic foci.

Three other patients needed revision surgery for persistent or recurrent seizures. Patients 14 and 16 had seizure recurrence with predominantly vegetative symptoms and had an amygdala residue resection: they are currently seizure-free [22]. 

Patient 24 (who was not considered in the analysis of long-term results) needed a second surgical intervention for persistent infantile spasm. Postoperative MRI showed a suspicion of a persistent callosal connection at the genu. He underwent a second surgery for completion of the callosotomy and amygdalohippocampectomy but, despite some seizure improvement, he was not seizure-free one month after this procedure. 

**Table 4 brainsci-13-01395-t004:** Comparison between our UCL results and published series of patients treated using vertical parasagittal hemispherotomy.

Author, yr	N	Etiologies (%)			Mean Age at		Complications										Mean FUp (yr)	Sz Outcome (Engel at Last FUp)							
		Cong	Acq	Prog	Onset (yr)	Surgery (yr)	Mty		HCP		BTF		APOS		Other			I		II		III		IV	
							n	%	n	%	n	%	n	%	n	%		n	%	n	%	n	%	n	%
Delalande, 2007 ¹ [17]	83	40 (48)	18 (21)	25 (30)	2.1	8.0	3	3.6	12	14.5	6	7.2	NA	NA	2	2.4	4.4	60	72.3	10	12.0	9	10.8	2	2.4
Honda, 2013 [23]	12	12 (100) ^2^	0	0	0.05	0.36	0	0.0	1	8.3	12	100.0	NA	NA	0	0.0	6.5	8	66.7	0	0.0	1	8.3	3	25.0
Dorfer, 2013 ^3^ [24]	37	13 (32)	26 (65)	1 (2)	1.2	5.5	1	2.7	1 ⁴	2.7	2	5.4	NA	NA	0	0.0	37	34	91.9	0	0.0	0	0.0	3	8.1
Kawai, 2014 [25]	7	4 (57)	3 (43)	0	2.1	14.8	0	0.0	0	0.0	NA	NA	0	0.0	0	0.0	3.1	6	85.7	0	0.0	0	0.0	1	14.3
Panigrahi, 2016 [26]	16	1 (6)	10 (62)	5 (31)	2.9	6.5	0	0.0	1	6.3	NA	NA	4	25.0	NA	NA	2.2	15	93.8	NA	NA	NA	NA	NA	NA
Fohlen, 2019 [27]	18	18 (100)	0	0	2	7.2	0	0.0	2	11.1	NA	NA	NA	NA	1	5.6	12.8	16	88.9	2	0.0	0	0.0	0	0.0
Saint-Luc, 2023	23	5	13	4	2.9	7.8	0	0.0	2	8.7	4	17.4	4	17.4	4 ⁵	17	6.81	20	86.9	2	8.7	0	0.0	1	4.3

Acq, acquired (infarction, postop/post-traumatic encephalomalacia, brain tumor); APOS, acute postoperative seizure; BTF, blood transfusion; Cong, congenital; Prog, progressive; NA, not available; VPS, ventriculoperitoneal shunt; yr, year; Mty, mortality; HCP, hydrocephalus; BTF, blood transfusion; FUp, follow-up; Sz, seizure. ^1^. Engel classification for 81 patients because of 2 postop deaths. ^2^. Hemimegalencephaly only. ^3^. Analysis included 37 of 40 patients with a follow-up of at least 12 months. ^4^. VPS 2.7%: Three patients with hydrocephalus needed a temporary external shunt but only one needed a VP shunt. ^5^. One cerebral abscess 6 months postop and 3 s surgeries for recurrence of epilepsy.

### 3.4. Cognitive Outcome

Six patients (26%) had significant cognitive impairment at their most recent follow-up, with no possibility of social integration. Patient 3 was bedridden since seizure recurrence 10 months after the first VPH with uncontrolled epilepsy, despite two VPH procedures for Rasmussen’s encephalitis. Patient 5 was able to walk but had severe pre-operative cognitive impairment due to severe head trauma responsible for the onset of his epilepsy. Patient 7 had severe cognitive impairment as part of extensive cortical dysplasia with refractory epilepsy and encephalopathy for 10 years prior to surgery. 

Patient 8 underwent surgery at the age of 22 with severe cognitive impairment present in the context of refractory epilepsy since birth. Patient 19 had global developmental delay, already suspected before surgery, in the context of his hemimegalencephaly, despite excellent control of his epilepsy. Finally, patient 20 had preoperative hemiplegia and severe preoperative congenital delay in the setting of a pre-natal left ischemic stroke. She is currently able to communicate but is unable to walk unassisted due to her persistent motor deficit. 

All the other patients were able to attend a normal school, or one adapted for their motor deficit. Patients 1 and 2 have a driver’s license and have completed university studies. Unfortunately, a quantitative assessment of the developmental quotients was not possible, due to the absence of pre- or postoperative assessment based on quantitative scales.

## 4. Discussion

### 4.1. Elegance of the Vertical Parasagittal Hemispherotomy (VPH) 

VPH allows complete disconnection of one hemisphere with minimal brain resection and without dissection of the Sylvian fissure, resection of the insula, or vessel sacrifice [15]. VPH provides complete insular cortex disconnection, avoiding potential poor postoperative outcomes due to a residual connected insular cortex as reported by Bulteau et al. [28] in cases of functional hemispherectomy. This complex procedure, involving one whole hemisphere, is carried out through a limited paramedian precentral cortical resection and through one ventricle with an extended view and understanding of the prosencephalic (forebrain) anatomy. 

### 4.2. Gelfoam Plug into the Foramen of Monro 

The range of postoperative shunt requirements varies between 2.5% and 23% [1,29]. In their series of 83 cases of VPH, Delalande et al. reported shunt placement in 16% [17], with an incidence of 76.9% in the hemimegalencephaly group with no clear explanation. In our three cases with hemimegalencephaly, no hydrocephalus developed. A similar rate of 13% has been reported with lateral hemispherotomy [30]. 

Hydrocephalus is considered a major complication after hemispherotomy. Acute hydrocephalus is associated with a risk of intracranial hypertension, cerebral herniation, and an altered level of consciousness, which requires urgent management by an external ventricular shunt. In chronic hydrocephalus with shunt dependence, the patient is exposed to the long-term disadvantages of a shunt (obstruction, disconnection, infection, hardware erosion, ascites). Avoidance of this complication is therefore a key factor in the management of these patients, in order to maximize their quality of life [1,31,32].

In both vertical and lateral approaches, shunt dependency is probably related to blood contamination of the CSF with inflammation of the subarachnoid spaces and arachnoid villi [33] and to the initial volume of the ventricular system (the smaller the ventricular system, the higher the risk of ventricular synechia). We favored the CSF blood contamination hypothesis and thus decided to protect the third ventricle and the rest of the ventricular system by occluding the homolateral foramen of Monro with a thin piece of Gelfoam sponge. This plug is removed at the very end of surgery when hemostasis is complete and the operative field is perfectly clean. This strategy may have played a role in reducing the hydrocephalus rate to 8.7%. Moreover, we had no cases of acute hydrocephalus, which is potentially more dangerous. It would be interesting to know whether this measure was used in other series reporting low rates of postoperative shunt.

### 4.3. Sub-Insular Disconnection with LN Preservation

The physiology of the LN, i.e., the putamen and the globus pallidum, is important in many aspects: motor adjustment and skill acquisition [34,35], habit memory [36], and cognitive functions [37]. Decreased N-acetyl aspartate/creatine ratios in both LN in patients with bipolar disorder were reported by Lai et al. [38], stressing the role of LN in mood regulation. Abnormalities of the LN microstructure in patients with schizophrenia have also been observed [39]. The LN has multiple connections not only with the cortex, which is disconnected by the VPH, but also with the thalamus, the habenula, and the brainstem. Keeping the LN as intact as possible preserves some of these connections and their roles. 

Nevertheless, the clinical impact of keeping the LN intact is currently unknown. Postoperative walking is irregularly reported in the literature, with rates between 33% and 100%, and a great variability depending on the etiology of the epilepsy and the length of follow-up [8,23,24,27]. At their most recent follow-up, 86.9% of our patients were able to walk independently; this proportion was 100% in those who could walk preoperatively. And 74% of patients were able to follow a conventional educational program with adaptations related to the loss of dexterity of the paretic hand. In patients with a bad functional status at the most recent follow-up, severe preoperative involvement, a longer duration of epilepsy, and underlying pathology (Rasmussen encephalopathy, hemimegalencephaly, etc.) are probably partly responsible for the poorer outcome [9,18]. Because of the small number of patients, it was not possible to demonstrate a statistically significant relationship between epilepsy etiology and functional outcome in our series. 

The importance of keeping the LN as intact as possible should be explored and analyzed on a larger series with long-term follow-up.

### 4.4. Acute Postoperative Seizures (APOSs) 

The definition of APOS has evolved over time and the influence of these seizures on prognosis is still unclear. In 1963, Falconer et al. reported ‘neighborhood’ fits in the first postoperative month without a worse prognosis [40]. From 1991, the time frame of 7 days was used and APOSs during this period were considered not predictive of prolonged seizure outcomes [41,42]. In 2001, the Commission on Neurosurgery of the ILAE recommended using a period of one month to define APOS, and seizures occurring during this period are not considered to have a negative prognostic value [43]. 

During the first postoperative month after surgery for DRE, 25% of children may have APOS [44]. In their series of patients treated using hemispherotomy, Panigrahi et al. [26] described a 25% rate of APOS, while their rate of seizure freedom (Engel I) at last follow-up was 94%, the highest in the literature. 

On the other hand, in their multicenter series, De Palma et al. [29] reported the presence of APOS as the only significant association with poor seizure outcome, and numerous studies have shown a correlation between early postoperative seizures and poor outcomes [45,46,47,48]. 

In our series, 66.6% of patients with APOS were seizure-free (Engel I) at the latest follow-up. There was no statistically significant correlation between the presence of APOS and the risk of not being seizure-free. However, because of the small number of patients, the results should be interpreted with caution.

In this context, we believe that caution be taken in the event of APOS. Surgery should not be immediately considered a failure, but patients/families should be warned that the surgical treatment may prove to be ineffective in one third of cases. 

### 4.5. VPH: Literature Review and Comparison with Our Series (Table 4) 

Including the present study, nine original cases series have been reported [17,23,24,25,26,27,29,49,50] in which the classical Delalande or modified VPH procedures were used, and seven papers presented enough data for comparison. 

Our series included predominantly acquired cases of DRE (55%) with an 86.9% rate of seizure freedom. By contrast, in a series by Honda et al. with only congenital cases, the seizure freedom rate was 66.7% [23].

The mean age at surgery in our series (7.3 y) is very similar to that in the population of Delalande et al. (8.0 y) [17]. The lowest mean age at surgery (4.3 y) was reported in the series by Honda et al. [23], probably because all their patients were congenital cases. In their series, all patients received a blood transfusion compared to 20% of our patients. In our population, the need for transfusion was significantly related to patient age (*p* = 0.0034) and our low rate of transfusion (20%) can then be explained by the small number of cases of hemimegalencephaly or Sturge–Weber syndrome. 

With 86.9% of patients in Engel class I at one-year follow-up and only one in Engel class IV (4%), our series confirms that sub-insular VPH is a very efficient procedure to treat refractory hemispheric DRE.

## 5. Limitations

Our study is limited by its observational nature. 

The lack of quantitative assessment of developmental quotients and the small number of patients prevents us from demonstrating a formal correlation between LN preservation and better cognitive results and does not enable a quantitative comparison with the literature.

## 6. Conclusions

The modified sub-insular VPH is a safe and successful surgical technique for hemispheric DRE with a similar seizure freedom rate to those reported in the largest series in the literature. Our modifications permitted a low rate of postoperative hydrocephalus and excellent motor and cognitive long-term outcomes. Longer follow-up periods and quantitative measurement of the pre- and postoperative cognitive status are needed to further assess the impact of the sub-insular approach.

## Figures and Tables

**Figure 2 brainsci-13-01395-f002:**
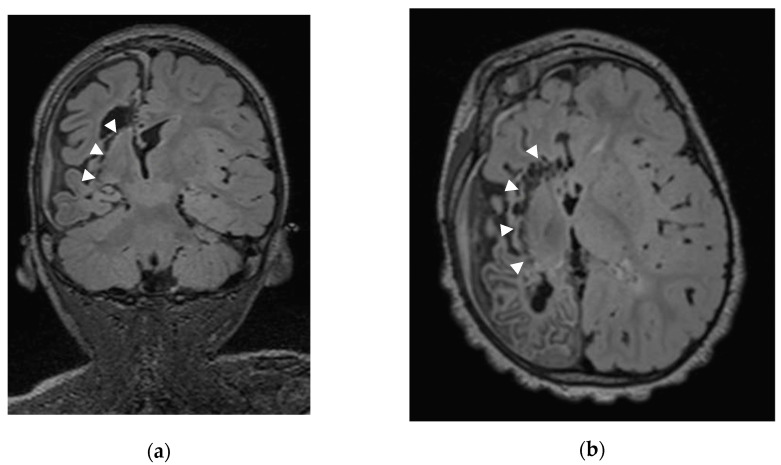
Coronal (**a**) and axial (**b**) MRI of Patient 21 (Sturge–Weber Syndrome) with arrows showing the sub-insular disconnection with preservation of the lenticular nucleus.

**Figure 3 brainsci-13-01395-f003:**
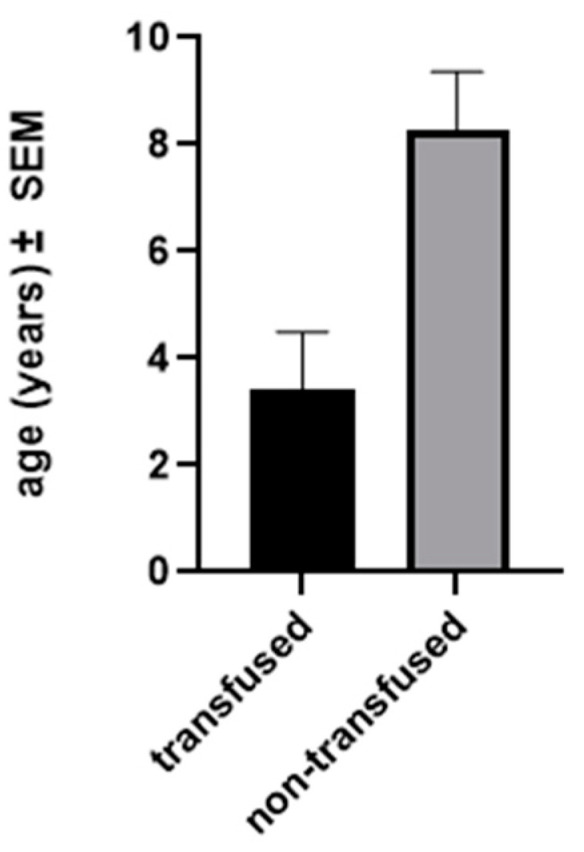
Comparison of age between transfused and non-transfused patients. Patients who were transfused during the operation were significantly younger than patients who were not transfused, *p* = 0.0154; SEM, standard error of the mean.

**Figure 4 brainsci-13-01395-f004:**
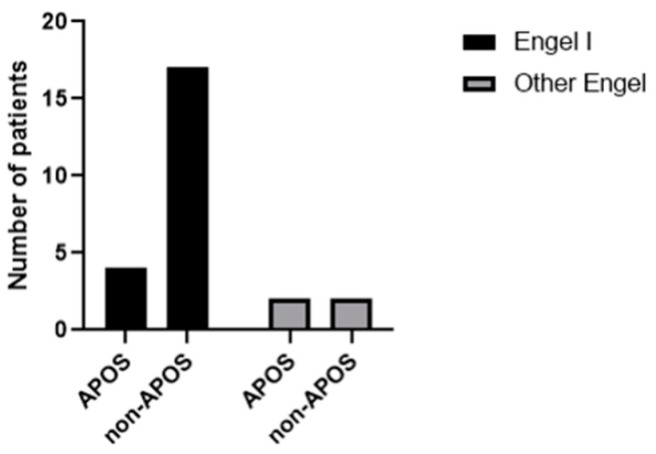
Comparison of epilepsy outcome between APOS and non-APOS patients. There was no statistically significant link between the presence of APOS and the risk of not being seizure-free. *p* = 0.234.

**Figure 5 brainsci-13-01395-f005:**
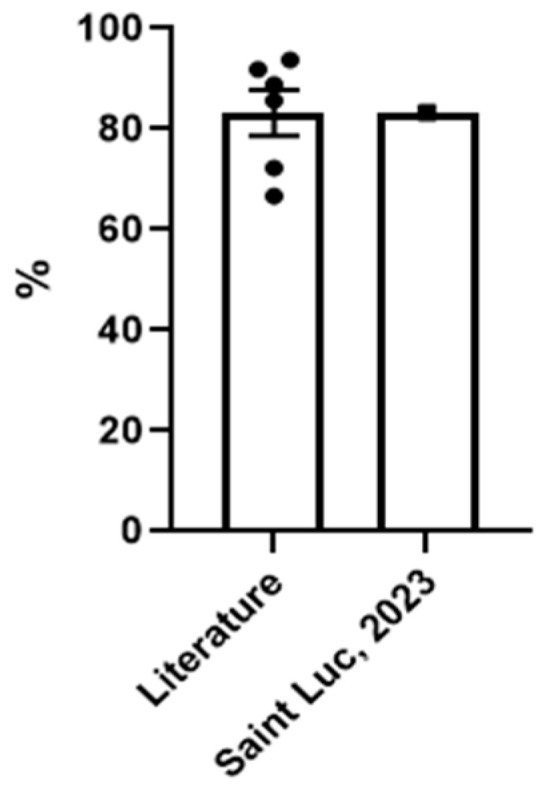
Comparison of proportion of patients with Engel score I (mean ±,SEM) at the final follow-up in the literature and in our Saint Luc University hospital case series. Our results are comparable with those reported in the literature for seizure outcome.

**Table 1 brainsci-13-01395-t001:** Comparison between classical Delalande and modified Raftopoulos sub-insular VPH.

	Delalande	Raftopoulos
Incision	1/3 anterior and 2/3 posterior to the coronal suture	Slightly anterior to the coronal suture
First step	Splenium disconnection	Anterior corpus callosotomy
Subrostral resection	Resection of the posterior part of the gyrus rectus	Resection of the posterior part of the gyrus rectus, of the cingulum, and of Brodmann area 25
Splenium disconnection	Until the roof of the third ventricle	Down to the great cerebral vein of Galen
Posterior disconnection	Posterior column of the fornix disconnection	Ventricular trigone floor (posterior column of fornix but also the intralimbic and limbic gyri)
Lateral disconnection	Lateral to the thalamus, going through the globus pallidus	Sub-insular trans-claustral
Temporal disconnection	Anterior part of the temporal horn resection	Piriform lobe resection

**Table 2 brainsci-13-01395-t002:** Preoperative data.

	Overall (25) No. of Patients (%) or Mean
Age at	
Onset	2.7
Surgery	7.3
Sex M/F	14/11
Etiology	
Congenital	6 (24%)
Cortical dysplasia	2 (8%)
Hemimegalencephaly	3 (12%)
Sturge–Weber	1 (4%)
Acquired	16 (64%)
Stroke (ischemic/hemorrhagic)	13 (52%)
Post-traumatic	1 (4%)
Gliosis (post-tumor resection)	1 (4%)
Progressive (Rasmussen)	4 (16%)
Hemispherotomy side (R/L)	17/8

## Data Availability

The data presented in this study are available in the article.

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
