# Peer review of "Modified Vertical Parasagittal Sub-Insular Hemispherotomy—Case Series and Technical Note"

_brainsci, 2023, doi:10.3390/brainsci13101395_

Round 1

Reviewer 1 Report

Dear Authors,

I have reviewed the manuscript and provided detailed feedback. Please refer to the attached file for the comprehensive review and specific suggestions for improvement.

Reviewer 2 Report

Major issues

#1. Introduction is too short. Since the authors want to describe a new method, please explain more about hemispherectomy and hemispherotomy. For example, the procedure difference etc.

#2. To easily understand your procedures, please show illustrations of VPH.

#3. I could not understand the procedure clearly. Do you want to say that adding the disconnection of the insula cortex or explain the new technique of hemispherotomy?

#4. Blood flow to the ventricles might not be related to hydrocephalus, but persistent high fevers according to a paper from epilepsy and behavior.

Minor issues

#1. In Abstract, 86,3% must be 86.3%

#2. In Introduction, DRE needs full words even though, it is shown in Abstract.

#3. It is unnatural to describe hemispherotomy as HST. Just use the word “hemispherotomy”, please.

There are errors, such as decimal points being represented as commas, suggesting it might not have been proofread by a professional. The content is also challenging to read. Would you consider having it checked by a professional English proofreader and obtaining their verification?

Round 2

Reviewer 1 Report

The paper underwent a thorough review, resulting in significant improvements. However, I have one correction to suggest for reference  41, from line 295 to 297 ''comes [39–40]. In 2001, the Commission on Neurosurgery of the ILAE recom- 295 mended using a period of one month to define APOS, and seizures occurring during this 296 period are not considered to have a negative prognostic value [42].'' as you can see 41 is missing please correct.

Reviewer 2 Report

Thak you for providing the new PDF attachment. I have reviewed it, and I believe that this figure is acceptable. I endorse this version, It is a good paper.